# CAUSALLY-GUIDED REGULARIZATION OF GRAPH ATTENTION IMPROVES GENERALIZABILITY

## ABSTRACT

Graph attention networks estimate the relational importance of node neighbors to aggregate relevant information over local neighborhoods for a prediction task. However, the inferred attentions are vulnerable to spurious correlations and connectivity in the training data, hampering the generalizability of models. We introduce CAR, a general-purpose regularization framework for graph attention networks. Embodying a causal inference approach based on invariance prediction, CAR aligns the attention mechanism with the causal effects of active interventions on graph connectivity in a scalable manner. CAR is compatible with a variety of graph attention architectures, and we show that it systematically improves generalizability on various node classification tasks. Our ablation studies indicate that CAR hones in on the aspects of graph structure most pertinent to the prediction (e.g., homophily), and does so more effectively than alternative approaches. Finally, we also show that CAR enhances interpretability of attention coefficients by accentuating node-neighbor relations that point to causal hypotheses.

## 1 INTRODUCTION

Graphs encode rich relational information that can be leveraged in learning tasks across a wide variety of domains. Graph neural networks (GNNs) can learn powerful node, edge or graph-level representations by aggregating a node's representations with that of its neighbors. The specifics of a GNN's neighborhood aggregation scheme are critical to its effectiveness on a prediction task. For instance, graph convolutional networks (GCNs) aggregate information via a simple averaging or max-pooling of neighbor features. GCNs are prone to suffer in many real-world scenarios where uninformative or noisy connections exist between nodes (Kipf & Welling, 2017; Hamilton et al., 2017). Graph-based attention mechanisms combat these issues by quantifying the relevance of node-neighbor relations and softly selecting neighbors in the aggregation step accordingly (Velickovic et al., 2018; Brody et al., 2022; Shi et al., 2021). This process of attending to select neighbors has contributed to significant performance gains for GNNs across a variety of tasks (Zhou et al., 2018; Veličković, 2022). Similar to the use of attention in natural language processing and computer vision, attention in graph settings also enables the interpretability of model predictions via the examination of attention coefficients (Serrano & Smith, 2019).

However, graph attention mechanisms can be prone to spurious edges and correlations that mislead them in how they attend to node neighbors, which manifests as a failure to generalize to unseen data (Knyazev et al., 2019). One approach to improve GNNs' generalizability is to regularize attention coefficients in order to make them more robust to spurious correlations/connections in the training data. Previous work has focused on $L_0$ regularization of attention coefficients to enforce sparsity (Ye & Ji, 2021) or has co-optimized a link prediction task using attention (Kim & Oh, 2021). Since these regularization strategies are formulated independently of the primary prediction task, they align the attention mechanism with some intrinsic property of the input graph without regard for the training objective.

We take a different approach and consider the question: "What is the importance of a specific edge to the prediction task?" Our answer comes from the perspective of regularization: we introduce CAR, a causal attention regularization framework that is broadly suitable for graph attention network architectures (**Figure** 1). Intuitively, an edge in the input graph is important to a prediction task if removing it leads to substantial degradation in the prediction performance of the GNN. The

key conceptual advance of this work is to scalably leverage active interventions on node neighborhoods (i.e., deletion of specific edges) to align graph attention training with the causal impact of these interventions on task performance. Theoretically, our approach is motivated by the invariant prediction framework for causal inference (Peters et al., 2016; Wu et al., 2022). While some efforts have previously been made to infuse notions of causality into GNNs, these causal approaches have been largely limited to using causal effects from pre-trained models as features for a separate model (Feng et al., 2021; Knyazev et al., 2019) or decoupling causal from non-causal effects (Sui et al., 2021).

We apply CAR on three graph attention architectures across eight node classification tasks, finding that it consistently improves test loss and accuracy. CAR is able to fine-tune graph attention by improving its alignment with task-specific homophily. Correspondingly, we found that as graph heterophily increases, the margin of CAR's outperformance widens. In contrast, a non-causal approach that directly regularizes with respect to label similarity generalizes less well. On the ogbn-arxiv network, we investigate the citations up/down-weighted by CAR and found them to broadly group into three intuitive themes. Our causal approach can thus enhance the interpretability of attention coefficients, and we provide a qualitative analysis of this improved interpretability. We also present preliminary results demonstrating the applicability of CAR to graph pruning tasks. Due to the size of industrially relevant graphs, it is common to use GCNs or sampling-based approaches on them. There, using attention coefficients learned by CAR on sampled subnetworks may guide graph rewiring of the full network to improve the results obtained with convolutional techniques.

## 2 METHODS

### 2.1 GRAPH ATTENTION NETWORKS

Attention mechanisms have been effectively used in many domains by enabling models to dynamically attend to the specific parts of an input that are relevant to a prediction task (Chaudhari et al., 2021). In graph settings, attention mechanisms compute the relevance of edges in the graph for a prediction task. A neighbor aggregation operator then uses this information to weight the contribution of each edge (Lee et al., 2019a; Li et al., 2016; Lee et al., 2019b).

The approach for computing attention is similar in many graph attention mechanisms. A graph attention layer takes as input a set of node features $\mathbf{h} = \{\vec{h}_1, ..., \vec{h}_N\}, \vec{h}_i \in \mathbb{R}^F$, where $N$ is the number of nodes. The graph attention layer uses these node features to compute attention coefficients for each edge: $\alpha_{ij} = a(\mathbf{W}\vec{h}_i, \mathbf{W}\vec{h}_j)$, where $a : \mathbb{R}^{F'} \times \mathbb{R}^{F'} \to (0, 1)$ is the attention mechanism function, and the attention coefficient $\alpha_{ij}$ for an edge indicates the importance of node $i$'s input features to node $j$. For a node $j$, these attention coefficients are then used to compute a linear combination of its neighbors' features: $\vec{h}'_j = \sum_{i \in N(j)} \alpha_{ij} \mathbf{W}\vec{h}_i, \quad s.t. \sum_{i \in N(j)} \alpha_{ij} = 1$. For multi-headed attention, each of the $K$ heads first independently calculates its own attention coefficients $\alpha_{i,j}^{(k)}$ with its head-specific attention mechanism $a^{(k)}(\cdot, \cdot)$, after which the head-specific outputs are averaged.

In this paper, we focus on three widely used graph attention architectures: the original graph attention network (GAT) (Velickovic et al., 2018), a modified version of this original network (GATv2) (Brody et al., 2022), and the Graph Transformer network (Shi et al., 2021). The three architectures and their equations for computing attention are presented in **Appendix** A.1.

### 2.2 CAUSAL ATTENTION REGULARIZATION: AN INVARIANCE PREDICTION FORMULATION

CAR is motivated by the invariant prediction (IP) formulation of causal inference (Peters et al., 2016; Wu et al., 2022). The central insight of this formulation is that, given sub-models that each contain a different set of predictor variables, the underlying causal model of a system is comprised of the set of all sub-models for which the predicted class distributions are equivalent, up to a noise term. This approach is capable of providing statistically rigorous estimates for both the causal effect strength of predictor variables as well as confidence intervals. With CAR, our core insight is that the graph structure itself, in addition to the set of node features, comprise the set of predictor variables. This is equivalent to the intuition that relevant edges for a particular task should not only be assigned high attention coefficients but also be important to the predictive accuracy of the model (**Figure** 1).

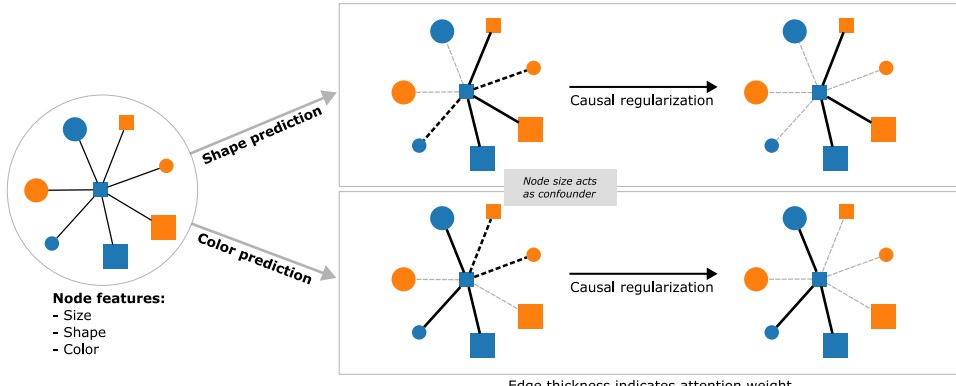

**Figure 1:** Schematic of CAR: Graph attention networks learn the relative importance of each node-neighbor for a given prediction task. However, their inferred attention coefficients can be miscalibrated due to noise, spurious correlations, or confounders (e.g., node size here). Our causal approach directly intervenes on a sampled subset of edges and supervises an auxiliary task that aligns an edge's causal importance to the task with its attention coefficient.

The removal of these relevant edges from the graph should cause the predictions that rely on them to substantially worsen.

We leverage the residual formulation of IP to formalize this intuition. This formulation assumes that we can generate sub-models for different sets of predictor variables, each corresponding to a separate experiment $e \in \mathcal{E}$. For each sub-model $S^e$, we compute the predictions $Y^e = g(\mathcal{G}^e_S, \mathcal{X}^e_S, \epsilon^e)$ where $\mathcal{G}$ is the graph structure, $\mathcal{X}$ is the set of features associated with $\mathcal{G}$, $\epsilon^e$ is the noise distribution, and $S$ is the set of predictor variables corresponding to $S^e$. We next compute the residuals $\mathcal{R} = Y - Y^e$. IP requires that we perform a hypothesis test on the means of the residuals, with the generic approach being to perform an F-test for each sub-model against the null-hypothesis. The relevant assumptions ($\epsilon^e \sim F_\epsilon$, and $\epsilon^e \perp\!\!\!\perp S^e$ for all $e \in \mathcal{E}$) are satisfied if and only if the conditionals $Y^e|S^e$ and $Y^f|S^f$ are identical for all experiments $e, f \in \mathcal{E}$.

We use an edge intervention-based strategy that corresponds precisely to this IP-based formulation. However, we differ from the standard IP formulation in how we estimate the final causal model. While IP provides a method to explicitly construct an estimator of the true causal model (by taking the intersection of all models for which the null hypothesis was rejected), we rely on intervention-guided regularization of graph attention coefficients as a way to aggregate sub-models while balancing model complexity and runtime considerations. In our setting, each sub-model corresponds to a set of edge interventions and, thus, slightly different graph structures. The same GNN architecture is trained on each of these sub-models. Given a set of experiments $\mathcal{E} = \{e\}$ with sub-models $S^e$, outputs $Y^e$ and errors $\epsilon^e$, we regularize the attention coefficients to align with sub-model errors, thus learning a GNN architecture primarily from causal sub-models. Incorporating this regularization as an auxiliary task, we seek to minimize the following loss:

$$\mathcal{L} = \mathcal{L}^p + \lambda \mathcal{L}^c \tag{1}$$

The full loss function $\mathcal{L}$ consists of the loss associated with the prediction $\mathcal{L}^p$, the loss associated with causal attention task $\mathcal{L}^c$, and a causal regularization strength hyperparameter $\lambda$ that mediates the contribution of the regularization loss to the objective. For the prediction loss, we have $\mathcal{L}^p = \frac{1}{N} \sum_{n=1}^{N} \ell^p(\hat{y}^{(n)}, y^{(n)})$, where $N$ is the size of the training set, $\ell^p(\cdot, \cdot)$ corresponds to the loss function for the given prediction task, $\hat{y}^{(n)}$ is the prediction for entity $n$, and $y^{(n)}$ is the ground truth value for entity $n$. We seek to align the attention coefficient for an edge with the causal effect of removing that edge through the use of the following loss function:

$$\mathcal{L}^c = \frac{1}{R} \sum_{r=1}^{R} \left( \frac{1}{|S^{(r)}|} \sum_{(n,i,j) \in S^{(r)}} \ell^c\left(\alpha_{ij}^{(n)}, c_{ij}^{(n)}\right) \right) \tag{2}$$

Here, $n$ represents a single entity for which we aim to make a prediction. For a node prediction task, the entity $n$ corresponds to a node, and in a graph prediction task, $n$ corresponds to a graph. In this

paper, we assume that all edges are directed and, if necessary, decompose an undirected edge into two directed edges. In each mini-batch, we generate $R$ separate sub-models, each of which consists of a set of edge interventions $S^{(r)}$, $r = 1, \ldots, R$. Each edge intervention in $S^{(r)}$ is represented by a set of tuples $(n, i, j)$ which denote a selected edge $(i, j)$ for an entity $n$. Note that in a node classification task, $n$ is the same as $j$ (i.e., the node with the incoming edge). More details for the edge intervention procedure and causal effect calculations can be found in the sections below. The causal effect $c_{ij}^{(n)}$ scores the impact of deleting edge $(i, j)$ through a likelihood ratio test. This causal effect is compared to the edge's attention coefficient $\alpha_{ij}^{(n)}$ via the loss function $\ell^c(\cdot, \cdot)$. A detailed algorithm for CAR is provided in **Appendix** A.2.

**Edge intervention procedure**  We sample a set of edges in each round $r$ such that the prediction for each entity will strictly be affected by at most one edge intervention in that round to ensure effect independence. For example, in a node classification task for a model with one GNN layer, a round of edge interventions entails removing only a single incoming edge for each node being classified. In the graph property prediction case, only one edge will be removed from each graph per round. Because a model with one GNN layer only aggregates information over a 1-hop neighborhood, the removal of each edge will only affect the model's prediction for that edge's target node. To select a set of edges in the $L$-layer GNN case, edges are sampled from the 1-hop neighborhood of each node being classified, and sampled edges that lie in more than one target nodes' $L$-hop neighborhood are removed from consideration as intervention candidates.

This edge intervention selection procedure is crucial, as it enables the causal effect of each selected intervention $c_{ij}^{(n)}$ on an entity $n$ to be calculated independently of other interventions on a graph. Moreover, by selecting only one intervention per entity within each round, we can parallelize the computation of these causal effects across all entities per round instead of iteratively evaluating just one intervention for the entire graph per entity, significantly aiding scalability.

**Calculating task-specific causal effects**  We quantify the causal effect of an intervention at edge $(i, j)$ on entity $n$ through the use of an approximate likelihood-ratio test

$$c_{ij}^{(n)} = \sigma\left(\left(\rho_{ij}^{(n)}\right)^{d(n)} - 1\right) \quad \text{where} \quad \rho_{ij}^{(n)} = \frac{\ell^p\left(\hat{y}_{\setminus ij}^{(n)}, y^{(n)}\right)}{\ell^p\left(\hat{y}^{(n)}, y^{(n)}\right)} \tag{3}$$

Here, $\hat{y}_{\setminus ij}^{(n)}$ is the prediction for entity $n$ upon removal of edge $(i, j)$. $d(n)$ represents the node degree for node classification, while it represents the number of edges in a graph for graph classification. Interventions will likely have smaller effects on higher degree nodes due to the increased likelihood of there being multiple edges that are relevant and beneficial for predictions for such nodes. In graph classification, graphs with many edges are likely to be less affected by interventions as well. Exponentiating the relative intervention effect $\rho_{ij}^{(n)}$ by $d(n)$ is intended to adjust for these biases. We have experimented both with and without raising $\rho$ to the factor of $d$, but have found empirically better results with. We do not have a rigorous explanation as to why, although we suspect the correlated nature of edges implies some amount of shrinkage is necessary.

The link function $\sigma : \mathbb{R} \to (0, 1)$ maps its input to the support of the distribution of attention coefficients. The predictions $\hat{y}^{(n)}$ and $\hat{y}_{\setminus ij}^{(n)}$ are generated from the graph attention network being trained with CAR. We emphasize that $\hat{y}^{(n)}$ and $\hat{y}_{\setminus ij}^{(n)}$ are both computed during the same training run, rather than in two separate runs.

**Scalability**  CAR increases the computational cost of training in two ways: (i) additional evaluations of the loss function due to the causal effect calculations, and (ii) searches to ensure independent interventions. CAR performs $\mathcal{O}(RN)$ interventions, where $R$ is the number of interventions per entity and $N$ is the number of entities in the training set. Because our edge intervention procedure ensures that the sampled interventions per round are independent, the causal effects of these interventions can be computed in parallel. In addition, if edge interventions were sampled uniformly across the graph, ensuring the independence each intervention would require a $L$-layer deep BFS that has time complexity $\mathcal{O}(b^L)$, resulting in a worst case time complexity of $\mathcal{O}(RNb^L)$, where $L$

is the depth of the GNN, and $b$ is the mean in-degree. We note that, in practice, GNNs usually have $L \leq 2$ layers since greater depth increases the computational cost and the risk of oversmoothing (Li et al., 2018; Chen et al., 2020; Topping et al., 2022). The contribution of this intervention overlap search step would, therefore, be minimal for most GNNs. We are also able to mitigate much of the computational cost by parallelizing these searches. Further speedups can achieved by identifying non-overlapping interventions as a preprocessing step. In summary, we have found CAR to have a modest effect on the scalability of graph attention methods, with training runtimes that are only increased by 1.5-2 fold (**Appendix** A.3).

## 3 RELATED WORK

The performance gains associated with graph attention networks have led to a number of efforts to enhance and better understand graph attention mechanisms (Lee et al., 2019a). One category of methods aims to improve the expressive power of graph attention mechanisms by supplementing a given prediction objective with a supervised attention mechanism (Feng et al., 2021), or a self-supervised connectivity prediction approach (Kim & Oh, 2021). A related set of methods leverage signals from interventions on graphs from a causal perspective to aid in training GNNs. One class of techniques performs interventions on nodes (Knyazev et al., 2019; Feng et al., 2021). CAL (Sui et al., 2021), a method designed for graph property prediction tasks, performs abstract interventions on representations of entire graphs instead of specific nodes or edges to identify the causally attended subgraph for a given task. Specifically, it uses these interventions to achieve robustness rather than directly leveraging information about the effect of these interventions on model predictions.

Our intervention-oriented framework can also be understood as a graph structure perturbation method. Perturbation methods can be broadly split into three different categories: graph data augmentation (Zhao et al., 2022), structural graph rewiring (Rong et al., 2020), and geometric graph rewiring (Topping et al., 2022). Inspired by the success of data augmentation approaches in computer vision, graph data augmentation methods seek to generate new training samples through different augmentation techniques. One of the earliest methods is DropEdge (Rong et al., 2020) reduces overfitting by randomly selecting edges from a uniform distribution to delete. Other methods build on DropEdge and select edges according to additional constraints including geometrical invariants (Gao et al., 2021), target probability distributions (Park et al., 2021), and information criteria (Suresh et al., 2021). EERM, a powerful invariance-based approach by Wu et al. (2022) takes a graph-editing approach to learn GNNs that are robust to distribution shifts in the data. Structural graph rewiring instead seeks to enforce structural priors such as sparsity or homophily during the graph alteration phase. Examples of these rewiring priors include fairness (Kose & Shen, 2022; Spinelli et al., 2022), temporal structure (Wang et al., 2021), predicted homophily (Chen et al., 2020), sparsity(Jin et al., 2020; Zheng et al., 2020), or information transfer efficiency (Klicpera et al., 2019). Geometric approaches, instead, choose to view the graph as a discrete geometry and alter the connectivity according to the balanced Forman curvature (Topping et al., 2022), the stochastic discrete Ricci flows (Bober et al., 2022), commute times (Arnaiz-Rodríguez et al., 2022), or algebraic connectivity(Arnaiz-Rodríguez et al., 2022). All of these approaches are designed either for use either in self-supervised learning or in a task-agnostic fashion, and consider the input graph independently of the task at hand.

To summarize, CAR introduces a combination of advances not previously reported: applicability to diverse attention architectures; task-based supervised regularization (rather than task-agnostic or self-supervised regularization) that leads to improved generalization; and a causal approach that scalably and directly relates an edge's importance to its attention coefficient, enhancing interpretability.

## 4 RESULTS

### 4.1 EXPERIMENTAL SETUP

We assessed the effectiveness of CAR by comparing the performance of a diverse range of models trained with and without CAR on 8 node classification datasets. Specifically, we aimed to assess the consistency of CAR's outperformance over matching baseline models across various graph attention mechanism and hyperparameter choices. Accordingly, we evaluated numerous combinations of such configurations (48 settings for each dataset and graph attention mechanism), rather than testing

only a limited set of optimized hyperparameter configurations. The configurable model design and hyperparameter choices that we evaluated include the graph attention mechanism (GAT, GATv2, or Graph Transformer), the number of graph attention layers $L = \{1, 2\}$, the number of attention heads $K = \{1, 3, 5\}$, the number of hidden dimensions $F' = \{10, 25, 100, 200\}$, and the regularization strength $\lambda = \{0.1, 0.5, 1, 5\}$. See **Appendix** A.4 for details on the network architecture, hyperparameters, and training configurations.

## 4.2 NODE CLASSIFICATION

**Datasets and evaluation:** We used a total of 8 real-world node classification datasets of varying sizes and degrees of homophily: Cora, CiteSeer, PubMed, ogbn-arxiv, Chameleon, Squirrel, Cornell and Wisconsin. Each model was evaluated according to its accuracy on a held-out test set. We also evaluated the test cross-entropy loss, as it accounts for the full distribution of per-class predictions rather than just the highest-valued prediction considered in accuracy calculations. See **Appendix** A.5 for details on dataset statistics, splits, and references.

**Generalization performance:** We compared both the test accuracy and the test loss of each model when trained with and without CAR. Across all model architecture and hyperparameter choices for a dataset, we applied the one-tailed paired Wilcoxon signed-rank test to quantify the overall outperformance of models trained with CAR against models trained without it. CAR resulted in higher test accuracy in 7 of the 8 node classification datasets and a lower test loss in all 8 datasets (**Figure** 2, $p < 0.05$). We report the relative performance when averaging over all hyperparameter choices in **Table** 1 (test loss), **Table** 7 (test accuracy, in **Appendix** A.6), and **Appendix** A.7. We also observed that even small values of $R$ were effective (**Appendix** A.8). We believe that this is due to there being an adequate number of causal-effect examples for regularization, as even in the minimum $R = 1$ case, we have roughly one causal-effect example per training example.

We also compared CAR with Sui et al. (2021)'s CAL, a method for graph property prediction that relies on an alternative formulation of causal attention with interventions on implicit representations. We adapted CAL for node classification by removing its final pooling layer. CAR-trained models substantially outperformed CAL (**Appendix** A.9), suggesting that CAR's direct edge-intervention approach results in better generalization. Taken together, these results highlight the consistency of performance gains achieved with CAR and its broad applicability across across graph attention architectures and hyperparameters.

**Table 1:** Test loss on 8 node classification datasets

|  | **Cora** | **CiteSeer** | **PubMed** | **ogbn-arxiv** | **Chameleon** | **Squirrel** | **Cornell** | **Wisconsin** |
|---|---|---|---|---|---|---|---|---|
| GAT | $2.33 \pm 0.75$ | $3.68 \pm 2.64$ | $1.05 \pm 0.24$ | $1.49 \pm 0.02$ | $1.29 \pm 0.03$ | $1.48 \pm 0.02$ | $1.19 \pm 0.23$ | $0.88 \pm 0.11$ |
| GAT + CAR | $\mathbf{1.82 \pm 0.77}$ | $\mathbf{3.06 \pm 2.34}$ | $\mathbf{0.89 \pm 0.18}$ | $1.49 \pm 0.02$ | $\mathbf{1.26 \pm 0.03}$ | $\mathbf{1.47 \pm 0.02}$ | $\mathbf{1.14 \pm 0.17}$ | $\mathbf{0.84 \pm 0.13}$ |
| GATv2 | $2.26 \pm 0.54$ | $3.08 \pm 1.06$ | $1.07 \pm 0.29$ | $1.48 \pm 0.02$ | $1.28 \pm 0.03$ | $1.48 \pm 0.02$ | $1.09 \pm 0.11$ | $0.88 \pm 0.15$ |
| GATv2 + CAR | $\mathbf{1.63 \pm 0.59}$ | $\mathbf{3.04 \pm 2.07}$ | $\mathbf{0.90 \pm 0.17}$ | $\mathbf{1.47 \pm 0.03}$ | $\mathbf{1.24 \pm 0.03}$ | $\mathbf{1.46 \pm 0.02}$ | $\mathbf{1.04 \pm 0.10}$ | $\mathbf{0.81 \pm 0.09}$ |
| Transformer | $3.43 \pm 1.61$ | $9.78 \pm 8.27$ | $1.79 \pm 0.88$ | $1.50 \pm 0.03$ | $1.38 \pm 0.06$ | $1.50 \pm 0.02$ | $1.22 \pm 0.25$ | $1.03 \pm 0.28$ |
| Transf. + CAR | $\mathbf{1.71 \pm 0.50}$ | $\mathbf{3.92 \pm 2.25}$ | $\mathbf{1.60 \pm 0.59}$ | $1.50 \pm 0.03$ | $\mathbf{1.33 \pm 0.04}$ | $\mathbf{1.48 \pm 0.03}$ | $\mathbf{1.07 \pm 0.12}$ | $\mathbf{0.86 \pm 0.13}$ |

## 4.3 MODEL INVESTIGATION AND ABLATION STUDIES

**Impact of regularization strength** We explored 4 settings of $\lambda$: $\{0.1, 0.5, 1, 5\}$. For 6 of the 8 node classification datasets, CAR models trained with the higher causal regularization strengths ($\lambda = \{1, 5\}$) demonstrated significantly larger reductions in test loss ($p < 0.05$, one-tailed Welch's $t$-test) compared to those trained with weaker regularization ($\lambda = \{0.1, 0.5\}$). Notably, all four of the datasets with lower homophily (Chamelon, Squirrel, Cornell and Wisconsin) displayed significantly larger reductions in test loss with the higher regularization strengths, suggesting that stronger regularization may contribute to improved generalization in such settings (**Appendix** A.10).

**Connection to spurious correlations and homophily** Graph attention networks that are prone to spurious correlations mistakenly attend to parts of the graph that are irrelevant to their prediction task. To evaluate if CAR reduces the impact of such spurious correlations, we assessed if models trained with CAR more effectively prioritized relevant edges. For node classification tasks, the relevant neighbors to a given node are expected to be those that share the same label as that node.

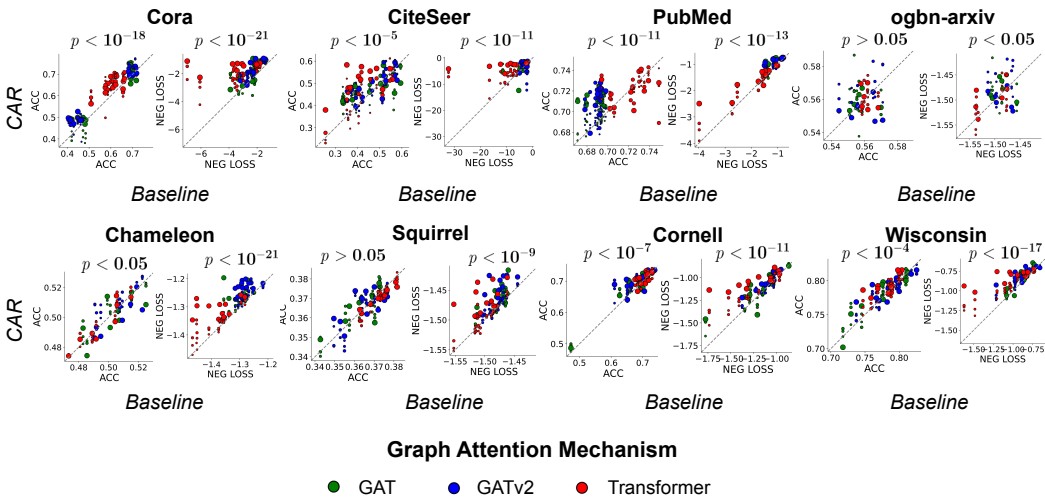

**Figure 2:** Test accuracy and negative loss on 8 node classification datasets. Each point corresponds to a comparison between a baseline model trained without CAR and an identical model trained with CAR. The point size represents the magnitude of the $\lambda$ value chosen for the CAR-trained model. $p$-values are computed from one-tailed paired Wilcoxon signed-rank tests evaluating the improvement of CAR-trained models over the baseline models.

We, therefore, used the label agreement between nodes connected by an edge as a proxy for the edge's ground-truth relevance. We assigned a reference attention coefficient $e_{ij}$ to an edge based on label agreement in the target node's neighborhood: $e_{ij} = \hat{e}_{ij} / \sum_{k \in N_j} \hat{e}_{kj}$ (Kim & Oh, 2021). Here, $\hat{e}_{ij} = 1$ if nodes $i$ and $j$ have the same label and $\hat{e}_{ij} = 0$ otherwise. $N_j$ denotes the in-neighbors of node $j$. We then calculated the KL divergence of an edge's attention coefficient $\alpha_{i,j}$ from its reference attention coefficient $e_{ij}$ and summarize a model's ability to identify relevant edges as the mean of these KL divergence values across the edges in the held-out test set.

We compared these mean KL divergences between baseline models trained without CAR and models trained with CAR across the same broad range of model architecture and hyperparameter choices described above. We found that CAR-trained models consistently yielded lower mean KL divergence values than models trained without CAR for 6 of 8 node classification datasets (**Figure** 3,$p < 0.05$, one-tailed paired Wilcoxon signed-rank test). Notably, this enhanced prioritization of relevant edges was achieved without explicitly optimizing label agreement during training and is an inherent manifestation of aligning attention with the node classification tasks' causal effects.

Low homophily graphs are associated with greater proportions of task-irrelevant edges and thus may introduce more spurious correlations (Zheng et al., 2022). We reasoned that CAR's relative effectiveness should be greater in such settings. We assessed this by evaluating CAR on 33 synthetic Cora datasets with varying levels of homophily (Zhu et al., 2020). We observed that CAR-trained models outperformed baseline models most substantially in low homophily settings, with performance gains generally increasing with decreasing homophily (**Appendix** A.11). Altogether, these results demonstrate that CAR not only more accurately prioritizes the edges that are most relevant to the desired task but also highlights its utility in low homophily settings most prone to spurious correlations.

**Comparison to homophily-based regularization** We next assessed if the performance gains by our causal approach could be replicated by a non-causal approach that systematically aligns attention coefficients with a generic measure of homophily. To do so, we performed an ablation study in which we replace the causal effects $c_{ij}^{(n)}$ computed from the network being trained (**Equation** 3) with an alternative score derived from a homophily-based classification scheme. Briefly, this scheme entails assigning a prediction for each node based on the counts of its neighbors' labels (see **Appendix** A.12 for details). By regularizing attention with respect to this homophily-based classification scheme, the attention mechanism for a network will be guided no longer by causal effects associated with the network but rather this separate measure of homophily.

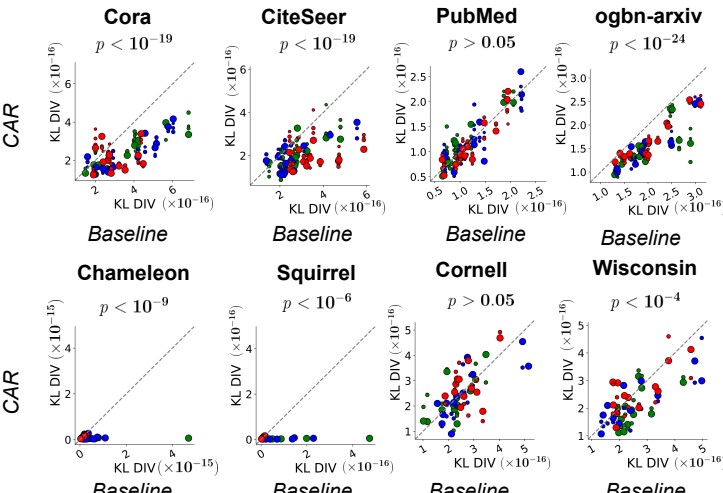

**Figure 3:** Coherence between attention coefficients and label agreement for CAR-trained models, compared to baseline. Lower KL divergence implies greater coherence. Point colors and sizes have the same meaning as in **Figure** 2. $p$-values are computed from one-tailed paired Wilcoxon signed-rank tests evaluating the improvement of CAR-trained models over the baseline models.

We compared CAR-trained models with those trained with homophily-based regularization by evaluating the consistency of their test loss improvements relative to the baseline models without regularization. We used the one-tailed paired Wilcoxon signed-rank test to evaluate the significance of the regularized models' test loss improvements across the set of model architecture and hyperparameter choices, focusing on models trained with the higher $\lambda \in \{1, 5\}$ regularization strengths. Models trained when regularizing attention coefficients with the homophily-based scheme underperformed those trained with CAR in 7 of the 8 datasets (**Table** 2). Interestingly, the homophily-based-based regularization showed an overall gain in performance relative to the baseline models (i.e., those trained without any regularization), suggesting that even non-specific regularization can be somewhat useful for training attention. Overall, these results demonstrate the effectiveness of using CAR to improve generalization performance for many node classification tasks.

**Table 2:** Ablation study comparing improvements in performance using CAR regularization and homophily-based regularization ($-\log_{10}(p)$, one-tailed paired Wilcoxon signed-rank test)

|            | Cora  | CiteSeer | PubMed | ogbn-arxiv | Chameleon | Squirrel | Cornell | Wisconsin |
|-----------:|-------|----------|--------|------------|-----------|----------|---------|-----------|
| CAR        | **12.36** | **5.92** | **10.51** | **1.81** | **12.77** | 9.72 | **6.72** | **9.34** |
| Homophily  | 10.04 | 5.59     | 7.30   | 0.53       | 12.34     | **10.97** | 4.15   | 8.20      |

**Additional applications** To explore the utility of CAR as a general-purpose framework, we employed CAR for graph pruning. CAR directly uses local pruning (i.e. edge interventions) to guide the training of graph attention in a manner that down-weights task-irrelevant edges. As such, we reasoned that attention coefficients produced by CAR-trained models could be used to prune task-irrelevant edges (see **Appendix** A.13 for more details). In this approach, we used CAR's edge pruning procedure as a pre-processing step for training and inference with GCNs, which are more scalable than graph attention networks (Rossi et al., 2020) but indiscriminate in how they aggregate information over node neighborhoods. We found that using CAR-guided pruning improved the test accuracy of GCNs, outperforming vanilla GCNs trained and evaluated over the full graph as well as GCNs trained and evaluated on graphs pruned with baseline graph attention mechanisms. These preliminary results open the door for further exploration of CAR's utility on these tasks.

## 4.4 INTERPRETING ATTENTION COEFFICIENTS: QUALITATIVE ANALYSIS

In addition to providing a robust way to increase both model quality and generalization, we explored the interpretability of CAR attention coefficients in a *post hoc* analysis. Here, we evaluated the edge-wise difference of the attention coefficients between our method and a baseline GAT applied to the ogbn-arxiv dataset. In this dataset, nodes represent arXiv papers, edges are citation links, and the prediction task is to classify papers into their subject areas.

We manually reviewed citations that were up/down-weighted by CAR-trained models and observed that these citations broadly fit into one of three categories: (i) down-weighting self-citations, (ii) down-weighting popular "anchor" papers, or (iii) upweighting topically-narrow papers with few citations. In the first case, we found that CAR down-weights edges associated with self-citations (**Table** 3). The causal story here is clear— machine learning is a fast-moving field with authors moving into the field and changing specialties as those specialties are born. Because of this, the narrative arc that a set of authors constructs to present this idea can include citations to their own previous work from different sub-fields. While these citations help situate the work within the broader literature and can provide background that readers might find valuable, they are not relevant to the subject area prediction task.

**Table 3:** Down-weighted self-citations.

| Paper Title | Cited Title (from a different subject area) | $\Delta$ |
|---|---|---|
| *Viterbinet a Deep Learning Based Viterbi Algorithm for Symbol Detection* | *Frequency Shift Filtering for Ofdm Signal Recovery in Narrowband Power Line Communications* | -89.91 % |
| *Solving Underdetermined Systems with Error Correcting Codes* | *Systems of MDS Codes from Units and Idempotents* | -89.80 % |
| *Performance Analysis for Multichannel Reception of OOFSK Signaling* | *On-Off Frequency-Shift Keying for Wideband Fading Channels* | -81.03 % |

In the second case, we found that CAR down-weights edges directed towards popular or otherwise seminal "anchor" papers (**Appendix** A.14, **Table** 11). These papers tends to be included in introductions to provide references for common concepts or methods, such as Adam, ResNet, and ImageNet. They are also widely cited across subject areas and hence have little bearing on the subject area prediction task. Notably, CAR does not simply learn to ignore edges from high-degree nodes. For the Word2Vec paper, we observed notable increases in attention coefficients for edges connecting it to multiple highly related papers, including a $2.5 \times 10^6$ % increase for *Efficient Graph Computation for Node2Vec* and a $2.0 \times 10^6$ % increase for *Multi-Dimensional Explanation of Reviews* .

In the final case, we observed that CAR up-weighted edges directed towards deeply related but largely unnoticed papers (**Appendix** A.14,**Table** 12). In our manual exploration of the data, we observed that these papers are those that are close to the proposed method. These papers are the type that are often found only after a thorough literature review. Such edges should play a key role in predicting a paper's topic and should be up-weighted.

## 5 CONCLUSION

We introduced CAR, an invariance principle-based causal regularization scheme that can be applied to graph attention architectures. Unlike other invariance-based approaches (Wu et al., 2022), our focus is on scalably improving overall generalization rather than handling distribution shifts. Towards that, we introduce an efficient scheme to directly intervene on multiple edges in parallel. Applying it to both homophilic and heterophilic node-classification tasks, we found accuracy improvements and loss reductions in almost all circumstances. We performed ablation studies for a deeper understanding, and found that CAR aligns attention with task-specific homophily and does so better than a homophily-based regularizer. A qualitative review also suggested that the attention-weight changes produced by CAR are intuitive and interpretable.

Understanding how, and improving what, GNNs learn remains a major open problem and is an active area of research. For instance, Zheng et al. (2022) have discussed the challenges that GNNs face when handling low-homophily graphs or when different tasks could be specified on the same underlying graph (e.g., predicting citation year vs. topic in obgn-arxiv). Towards this, our method provides a principled and scalable approach to align attention coefficients with the relevant task. Our work bridges two families of techniques: attention regularization and causal interventions. The synthesis of these techniques is not only a promising direction for enhancing the performance and interpretability of graph attention but also opens the door for leveraging similar techniques for general GNNs without attention as well. Lastly, while our graph pruning results are preliminary, they also suggest a promising direction for future work on scaling CAR-based insights to web-scale graphs.

## 6 REPRODUCIBILITY STATEMENT

To ensure the reproducibility of the results in this paper, we have included the source code for our method as supplementary materials. The datasets used in this paper are all publicly available, and we also use the publicly available train/validation/test splits for these datasets. We provide details on these datasets in the Appendix and have provided references to them in both the main text and the Appendix. In addition, we have provided detailed descriptions of the experimental setup, model training schemes, model architecture design choices, and hyperparameter choices in the "Experimental Setup" section as well as in Appendix A.4.

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

## A  Appendix

### A.1  Graph Attention Architecture Variants

**Table 4:** Attention coefficient calculation across graph attention architecture variants

| | |
|---|---|
| GAT (Velickovic et al., 2018) | $e_{ij} = \text{LeakyReLU}\big(\vec{\mathbf{a}}^T[\mathbf{W}\vec{h_i}\|\mathbf{W}\vec{h_j}]\big)$ |
| GATv2 (Brody et al., 2022) | $e_{ij} = \vec{\mathbf{a}}^T\text{LeakyReLU}\big(\mathbf{W}\vec{h_i}\|\mathbf{W}\vec{h_j}\big)$ |
| Graph Transformer (Shi et al., 2021) | $e_{ij} = \dfrac{(\mathbf{W}\vec{h_i})^T(\mathbf{W}\vec{h_j})}{\sqrt{F'}}$ |

### A.2  CAR Algorithm

---
**Algorithm 1** CAR Framework

---
**Input:** Training set $\mathcal{D}_{train}$, validation set $\mathcal{D}_{val}$, model $\mathcal{M}$, regularization strength $\lambda$
  **repeat**
    **for each** mini-batch $\{\mathcal{B}^k = \{n_j^{(k)}\}_{j=1}^{b_k}\}$ **do**
      Prediction loss: $\mathcal{L}^p \leftarrow \frac{1}{|\mathcal{B}^k|}\sum_{n\in\mathcal{B}^k}\ell^p(\hat{y}^{(n)}, y^{(n)})$
      **procedure** EDGE INTERVENTION
        Causal attention loss: $\mathcal{L}^c \leftarrow 0$
        **for** round $r \leftarrow 1$ to $R$ **do**
          Set of edge interventions $S(r) \leftarrow \{\}$
          **for each** entity $\{n_j^{(k)}\}_{j=1}^{b_k}$ **do**
            Sample edge $(i,j) \sim E_{n_j^{(k)}}$       $\triangleright E_{n_j^{(k)}}$ set of edges related to entity $n_j^{(k)}$
            **if** $(i,j)$ independent of $S(r)$ **then**     $\triangleright$ See "Edge intervention procedure"
              $S(r) \leftarrow S(r) \cup (n_j^{(k)}, i, j)$    $\triangleright$ Add edge to set of edge interventions
              Compute causal effect $c_{ij}^{(n)} \leftarrow \sigma\big((\rho_{ij}^{(n)})^{d(n)} - 1\big)$    $\triangleright$ Equation 5
            **end if**
          **end for**
          $\mathcal{L}^c \leftarrow \mathcal{L}^c + \frac{1}{R}\frac{1}{|S^{(r)}|}\sum_{(n,i,j)\in S^{(r)}}\ell^c\left(\alpha_{ij}^{(n)}, c_{ij}^{(n)}\right)$    $\triangleright$ Equation 3
        **end for**
      **end procedure**
      Total loss: $\mathcal{L} = \mathcal{L}^p + \lambda\mathcal{L}^c$
      Update model parameters to minimize $\mathcal{L}$
    **end for**
  **until** Convergence criterion     $\triangleright$ We use convergence of the validation prediction loss.

---

### A.3  Runtime Statistics

**Table 5:** Training times for node classification datasets

| Dataset | # nodes (train set) | # edges (train set) | Train time w/o CAR (sec) | Train time w/ CAR (sec) |
|---|---|---|---|---|
| Cora | 140 | 638 | $1.1 \pm 0.4$ | $2.2 \pm 2.1$ |
| CiteSeer | 120 | 364 | $1.3 \pm 0.5$ | $2.1 \pm 1.7$ |
| PubMed | 60 | 297 | $1.4 \pm 0.7$ | $2.3 \pm 2.6$ |
| ogbn-arxiv | 32,970 | 93,942 | $165 \pm 78$ | $286 \pm 166$ |
| Chameleon | 1,092 | $17,157 \pm 1,013$ | $1.6 \pm 0.3$ | $2.6 \pm 1.1$ |
| Squirrel | 2,496 | $105,517 \pm 3,062$ | $2.7 \pm 0.6$ | $4.0 \pm 1.9$ |
| Cornell | 87 | $148 \pm 18$ | $1.3 \pm 1.2$ | $1.4 \pm 0.9$ |
| Wisconsin | 120 | $239 \pm 10$ | $1.1 \pm 0.4$ | $1.1 \pm 0.6$ |

## A.4    Graph Attention Network Architectures and Training

### A.4.1    Node Classification Model Architecture

The GNN model used for node classification tasks takes as input the original node features $\vec{x}_i \in \mathbb{R}^d$ and applies a non-linear projection to these features to yield a set of hidden features $\vec{h}_i = \text{LeakyReLU}(\mathbf{W}_1 \vec{x}_i + \mathbf{b}_1)$, where $\mathbf{W}_1 \in \mathbb{R}^{F \times d}$ and $\mathbf{b}_1 \in \mathbb{R}^F$. These hidden features are then passed through $L$ graph attentional layers of the same chosen architecture, yielding new hidden features per node of the same dimensionality $\vec{h}'_i \in \mathbb{R}^F$. The pre- and post-graph attention layer hidden features are then concatenated $[\vec{h}_i || \vec{h}'_i]$, after which a final linear layer and softmax transformation $\sigma_{\text{softmax}}(\cdot)$ are applied to produce the prediction output $\hat{y}_i = \sigma_{\text{softmax}}(\mathbf{W}_2 \text{LeakyReLU}([\vec{h}_i || \vec{h}'_i]) + \mathbf{b}_2)$. Here, $\hat{y}_i \in \mathbb{R}^C$, $\mathbf{W}_2 \in \mathbb{R}^{2F \times C}$, and $\mathbf{b}_2 \in \mathbb{R}^F$, where $C$ is the number of classes in the classification task. Models were implemented in PyTorch and PyTorch Geometric (Fey & Lenssen, 2019). Self-loops were not included in the graph attention layers; otherwise, default PyTorch Geometric parameter settings were used for the graph attention layers.

### A.4.2    Training Details

We used cross-entropy loss for the prediction loss $\ell^p(\cdot, \cdot)$ and binary cross-entropy loss for the causal regularization loss $\ell^c(\cdot, \cdot)$. The link function $\sigma(\cdot)$ was chosen to be the sigmoid function with temperature $T = 0.1$. Unless otherwise specified, we performed $R = 5$ rounds of edge interventions per mini-batch when training with CAR.

All models were trained using the Adam optimizer with a learning rate of 0.01 and mini-batch size of 10,000. Each dataset was partitioned into training, validation, and test splits in line with previous work (**Appendix** A.5), and early stopping was applied during training with respect to the validation loss. Training was performed on a single NVIDIA Tesla T4 GPU.

## A.5    Datasets

We provide overviews of the various node classification datasets along with accompanying statistics in Table 6. For all datasets, we use the publically available train/validation/test splits that accompany these datasets.

**Planetoid:** The Cora, CiteSeer, and PubMed datasets are citation networks from Yang et al. (2016). Nodes represent documents and directed edges represent citation links. Nodes are featurized as bag-of-word representations of their respective documents. The prediction task for this dataset is to classify a given paper into its respective subject area.

**ogbn-arxiv:** The ogbn-arxiv dataset is a citation network between computer science arXiv paper indexed by MAG (Hu et al., 2020). Nodes represent papers and a node's features are the mean embeddings of words in its corresponding paper's title and abstract. Edges are directed and represent a citation by one paper of another. The prediction task for this dataset is to predict the subject area of a given arXiv paper.

**Wikipedia:** The Chameleon and Squirrel datasets are Wikipedia networks from Rozemberczki et al. (2021), in which nodes represent web pages and edges represent hyperlinks between them. Nodes are featurized as bag-of-word representations of important nouns in their respective Wikipedia pages. Average monthly traffic of web pages are converted into categories, and the prediction task is to assign a given page to its corresponding category.

**WebKB:** The Cornell and Wisconsin datasets are networks of web pages from various computer science departments, in which nodes represent web pages and edges are hyperlinks between them. Node features are bag-of-word representations of their respective web pages, and the prediction task is to assign a given web page to the category that describes its content.

**Table 6:** Node Classification Dataset Statistics

| Dataset | # classes | # nodes | # edges | # splits | Mean degree | Homophily |
|---|---|---|---|---|---|---|
| Cora | 7 | 2,708 | 10,556 | 1 | 3.9 | 0.81 |
| CiteSeer | 6 | 3,327 | 9,104 | 1 | 2.7 | 0.74 |
| PubMed | 3 | 19,717 | 88,648 | 1 | 4.5 | 0.80 |
| ogbn-arxiv | 40 | 169,343 | 1,166,243 | 1 | 6.9 | 0.66 |
| Chameleon | 5 | 2,277 | 36,051 | 10 | 15.8 | 0.23 |
| Squirrel | 5 | 5,201 | 216,933 | 10 | 41.7 | 0.22 |
| Cornell | 5 | 183 | 295 | 10 | 1.6 | 0.12 |
| Wisconsin | 5 | 251 | 499 | 10 | 2.0 | 0.17 |

## A.6 TEST ACCURACY ON NODE CLASSIFICATION DATASETS

**Table 7:** Test accuracy on 8 node classification datasets

| | Cora | CiteSeer | PubMed | ogbn-arxiv | Chameleon | Squirrel | Cornell | Wisconsin |
|---|---|---|---|---|---|---|---|---|
| GAT | $0.58 \pm 0.13$ | $0.46 \pm 0.09$ | $0.69 \pm 0.01$ | $0.56 \pm 0.01$ | $0.50 \pm 0.01$ | $0.37 \pm 0.01$ | $0.66 \pm 0.06$ | $0.78 \pm 0.03$ |
| GAT + CAR | $\mathbf{0.60 \pm 0.12}$ | $\mathbf{0.49 \pm 0.06}$ | $\mathbf{0.71 \pm 0.01}$ | $0.56 \pm 0.01$ | $0.50 \pm 0.01$ | $0.37 \pm 0.01$ | $\mathbf{0.67 \pm 0.06}$ | $0.78 \pm 0.03$ |
| GATv2 | $0.60 \pm 0.14$ | $0.47 \pm 0.08$ | $0.69 \pm 0.01$ | $0.56 \pm 0.01$ | $0.51 \pm 0.01$ | $0.37 \pm 0.01$ | $0.68 \pm 0.04$ | $0.79 \pm 0.02$ |
| GATv2 + CAR | $\mathbf{0.61 \pm 0.12}$ | $\mathbf{0.50 \pm 0.06}$ | $\mathbf{0.71 \pm 0.01}$ | $0.56 \pm 0.01$ | $0.51 \pm 0.01$ | $0.37 \pm 0.01$ | $\mathbf{0.70 \pm 0.03}$ | $0.79 \pm 0.02$ |
| Transformer | $0.60 \pm 0.04$ | $0.43 \pm 0.09$ | $0.72 \pm 0.02$ | $0.56 \pm 0.01$ | $0.50 \pm 0.01$ | $0.37 \pm 0.01$ | $0.70 \pm 0.02$ | $0.78 \pm 0.02$ |
| Transformer + CAR | $\mathbf{0.65 \pm 0.04}$ | $\mathbf{0.47 \pm 0.09}$ | $0.72 \pm 0.01$ | $0.56 \pm 0.01$ | $0.50 \pm 0.01$ | $0.37 \pm 0.01$ | $\mathbf{0.71 \pm 0.02}$ | $\mathbf{0.79 \pm 0.01}$ |

## A.7 CHANGES IN TEST ACCURACY AND LOSS BY GRAPH ATTENTION MECHANISM AND REGULARIZATION STRENGTH

**Table 8:** Average percent change in test accuracy

| $\lambda$ Value | Cora | | | CiteSeer | | | PubMed | | |
|---|---|---|---|---|---|---|---|---|---|
| | GAT | GATv2 | Transformer | GAT | GATv2 | Transformer | GAT | GATv2 | Transformer |
| $\lambda \in \{1.0, 5.0\}$ | **+3.8%** | **+7.4%** | +7.0% | **+7.4%** | **+7.1%** | **+12.5%** | **+2.7%** | +0.3% | 0.0% |
| $\lambda \in \{0.1, 0.5\}$ | +2.3% | +6.6% | **+7.6%** | +4.2% | +3.0% | +1.5% | +0.4% | +0.3% | 0.0% |

| $\lambda$ Value | ogbn-arxiv | | | Chameleon | | | Squirrel | | |
|---|---|---|---|---|---|---|---|---|---|
| | GAT | GATv2 | Transformer | GAT | GATv2 | Transformer | GAT | GATv2 | Transformer |
| $\lambda \in \{1.0, 5.0\}$ | -0.9% | -0.6% | **-0.3%** | -0.2% | -1.1% | -0.2% | +0.3% | **+0.4%** | -0.5% |
| $\lambda \in \{0.1, 0.5\}$ | **0%** | **-0.3%** | -0.7% | **+0.3%** | **+0.2%** | **0.0%** | **+0.5%** | -0.1% | **-0.4%** |

| $\lambda$ Value | Cornell | | | Wisconsin | | |
|---|---|---|---|---|---|---|
| | GAT | GATv2 | Transformer | GAT | GATv2 | Transformer |
| $\lambda \in \{1.0, 5.0\}$ | **+1.8%** | **+3.1%** | +0.6% | **+0.6%** | **+0.4%** | **+1.2%** |
| $\lambda \in \{0.1, 0.5\}$ | +1.4% | +1.6% | **+0.7%** | +0.4% | 0.0% | +0.9% |

**Table 9:** Average percent change in test loss

| $\lambda$ Value | Cora | | | CiteSeer | | | PubMed | | |
|---|---|---|---|---|---|---|---|---|---|
| | GAT | GATv2 | Transformer | GAT | GATv2 | Transformer | GAT | GATv2 | Transformer |
| $\lambda \in \{1.0, 5.0\}$ | **-22.6%** | **-29.0%** | **-43.6%** | -11.7% | **+7.6%** | **-45.0%** | **-12.8%** | **-13.9%** | **-7.0%** |
| $\lambda \in \{0.1, 0.5\}$ | -15.3% | -19.8% | -28.8% | **-12.9%** | +11.2% | -20.9% | -2.8% | -2.9% | -1.1% |

| $\lambda$ Value | ogbn-arxiv | | | Chameleon | | | Squirrel | | |
|---|---|---|---|---|---|---|---|---|---|
| | GAT | GATv2 | Transformer | GAT | GATv2 | Transformer | GAT | GATv2 | Transformer |
| $\lambda \in \{1.0, 5.0\}$ | +0.6% | +1.9% | -0.2% | **-2.4%** | **-2.7%** | **-3.8%** | **-1.1%** | **-1.5%** | **-2.0%** |
| $\lambda \in \{0.1, 0.5\}$ | **-0.1%** | **-0.2%** | **-0.8%** | -0.7% | -0.7% | -0.8% | -0.2% | 0.0% | 0.0% |

| $\lambda$ Value | Cornell | | | Wisconsin | | |
|---|---|---|---|---|---|---|
| | GAT | GATv2 | Transformer | GAT | GATv2 | Transformer |
| $\lambda \in \{1.0, 5.0\}$ | **-1.8%** | **-3.1%** | **-0.6%** | -0.6% | **-0.4%** | **-1.2%** |
| $\lambda \in \{0.1, 0.5\}$ | -1.4% | -1.6% | **-0.7%** | -0.4% | 0.0% | -0.9% |

### A.8 NUMBER OF ROUNDS OF INTERVENTIONS AND PERCENT CHANGE IN TEST ACCURACY

**Figure 4:** Percent change in test accuracy for models trained with CAR. Each boxplot represents all combinations of the three graph attention layers (GAT, GATv2, Transformer) and the following sets of hyperparameter choices: $\lambda \in \{1, 5\}$, $L = \{1, 2\}$, $K = \{3\}$, $F' = \{100, 200\}$.

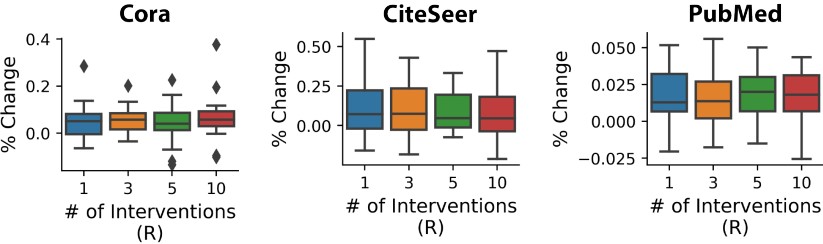

### A.9 COMPARISON OF CAR-TRAINED MODELS WITH CAL MODELS

CAL is an approach for identifying causally attended subgraphs for graph prediction tasks that leverages causal interventions on graph representations to achieve robustness of model predictions (Zhao et al., 2021). While CAL and CAR have related goals of enhancing graph attention using concepts from causal theory, CAL uses abstract perturbations on graph representation to perform causal interventions while we propose an edge intervention strategy that enables causal effects to be computed scalably. In addition, CAL is designed to identify causally attended subgraphs for graph property prediction tasks, while our work primarily focuses on node classification tasks. Furthermore, CAL uses interventions to achieve robustness and does not directly leverage the effects of interventions on model predictions during training.

Despite these differences, we sought to determine whether the causal principles underlying CAL could be effectively applied to the various node classification tasks evaluated in our paper. We modified the CAL architecture to make it suitable for node prediction tasks by simply removing the final pooling layer that aggregates node representations within each graph directly upstream of a classifier, thus enabling node-level prediction. We evaluated the CausalGAT model from CAL using all combinations of the following hyperparameter choices: $F' = \{128, 256\}$, $K = \{1, 2, 4\}$, $L = \{1, 2\}$, $\lambda_1 = \{0.2, 0.4, 0.6, 0.8, 1\}$, and $\lambda_2 = \{0.2, 0.4, 0.6, 0.8, 1\}$, where $F'$ refers to the number of hidden dimensions, $K$ is the number of attention heads, $L$ is the number of GNN layers, and $\lambda_1$ and $\lambda_2$ are CAL-specific hyperparameters. For each dataset, we report the maximum test accuracy observed for the CAL CausalGAT across all combinations of these hyperparameter choices. We compare these test accuracies from the CAL CausalGAT models with the test accuracies from CAR-trained models averaged over all hyperparameter choices, which also appear above in **Appendix A.6**.

**Table 10:** Test accuracy on 8 node classification datasets compared to CAL

|  | Cora | CiteSeer | PubMed | ogbn-arxiv | Chameleon | Squirrel | Cornell | Wisconsin |
|---|---|---|---|---|---|---|---|---|
| GAT + CAR | $0.60 \pm 0.12$ | $0.49 \pm 0.06$ | $0.71 \pm 0.01$ | $0.56 \pm 0.01$ | $0.50 \pm 0.01$ | $0.37 \pm 0.01$ | $0.67 \pm 0.06$ | $0.78 \pm 0.03$ |
| GATv2 + CAR | $0.61 \pm 0.12$ | $0.50 \pm 0.06$ | $0.71 \pm 0.01$ | $0.56 \pm 0.01$ | $0.51 \pm 0.01$ | $0.37 \pm 0.01$ | $0.70 \pm 0.03$ | $0.79 \pm 0.02$ |
| Transformer + CAR | $0.65 \pm 0.04$ | $0.47 \pm 0.09$ | $0.72 \pm 0.01$ | $0.56 \pm 0.01$ | $0.50 \pm 0.01$ | $0.37 \pm 0.01$ | $0.71 \pm 0.02$ | $0.79 \pm 0.01$ |
| CAL CausalGAT | 0.49 (best) | 0.37 (best) | 0.55 (best) | 0.21 (best) | 0.29 (best) | 0.31 (best) | 0.51 (best) | 0.47 (best) |

## A.10 REGULARIZATION STRENGTH AND GENERALIZATION PERFORMANCE

**Figure 5:** Test loss reduction for CAR-trained models across regularization strengths. $p$-values are computed from one-tailed $t$-tests evaluating the significance of the test loss reductions for the $\lambda \in \{1, 5\}$ CAR-trained models being greater than those of the $\lambda \in \{0.1, 0.5\}$ CAR-trained models.

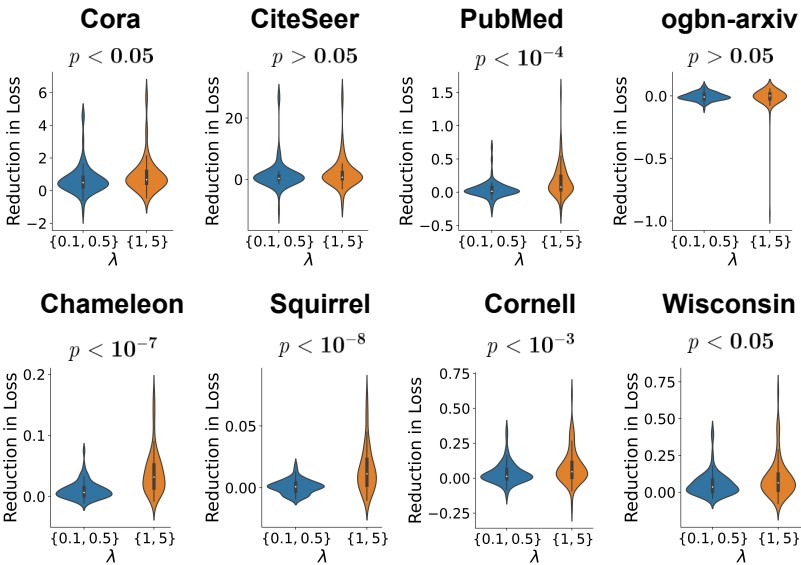

## A.11 EXPERIMENTAL SETUP AND TEST ACCURACY FOR SYNTHETIC CORA DATASETS

To assess the relationship between the effectiveness of CAR and the homophily of a dataset, we obtained a set of synthetic Cora datasets from (Zhu et al., 2020). These synthetic datasets are modified versions of the original Cora dataset that feature varying levels of edge homophily $h$, which is defined as the fraction of edges in a graph which connect nodes that have the same class label. Here, $\mathcal{E}$ is the set of edges, $y_u$ is the class label for node $u$, and $y_v$ is the class label for node $v$.

$$h = \frac{|\{(u, v) : (u, v) \in \mathcal{E} \wedge y_u = y_v\}|}{|\mathcal{E}|} \tag{4}$$

We evaluated 33 synthetic Cora datasets that spanned 11 different settings for $h$, each of which were represented by 3 replicate datasets. For each of these datsets, we performed a similar analysis as above, in which we aimed to evaluate the consistency of improvements in test loss using CAR across a number of graph attention and hyperparameter choices. We evaluated the GAT, GATv2, and Transformer graph attention layers along with all combinations of the following sets of hyperparameter choices: $F' = \{100\}$, $\lambda = \{1, 5\}$, $K = \{1, 3, 5\}$, $L = \{1, 2\}$. We then performed a one-tailed paired Wilcoxon rank-sum test to quantify the consistency of CAR-trained models' improvement in test loss over baseline models trained without CAR.

**Figure 6:** Generalization performance of CAR-trained models compared to baseline across various levels of edge homophily ($-\log_{10}(p)$, one-tailed paired Wilcoxon rank-sum test).

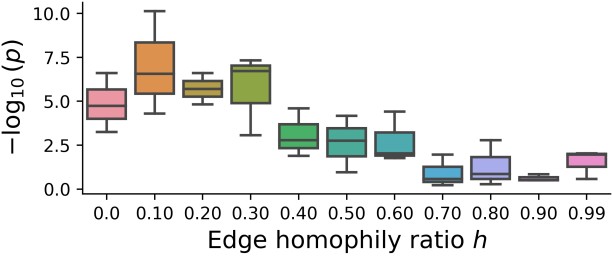

## A.12    HOMOPHILY-BASED REGULARIZATION DETAILS

For the neighbor voting model, each node's prediction is a softmax-normalized histogram of class labels from the node's neighbors: $\hat{y}^{(n)} = (p_1, \ldots, p_C)$, where $p_k$ is the normalized count of neighbors with label $k$. We employed the above edge intervention procedure and causal effect calculations to compute causal effect scores for interventions to this neighbor voting model $\tilde{c}_{ij}^{(n)}$. These values are used in place of the network-specific $c_{ij}^{(n)}$ values from the original implementation of CAR. Otherwise, the training procedure for a network trained with this neighbor voting scheme is exactly the same as for training with CAR. We note that, for a given node, calculating the intervention-affected prediction $\hat{y}_{\backslash ij}^{(n)}$ simply entails updating the normalized counts of the class labels from the node's remaining neighbors after the intervention.

## A.13    CAR-GUIDED GRAPH REWIRING

While graph attention networks have demonstrated notable performance gains, its inclusion of graph attention layers currently limits its use in large-scale applications compared to GCNs, for which a number of advances in scalability have been made (Rossi et al., 2020).

To leverage the advantages of CAR in graph attention alongside the scalability of GCNs, we explored a graph rewiring approach based on CAR-guided edge pruning. For a given dataset, we first use a trained graph attention network to assign an attention weight for each edge in the training and validation sets, after which edges with attention weights below a threshold $\alpha_T$ are pruned. A GCN is then trained on the rewired training set with early stopping imposed with respect to the validation loss on the rewired validation set. The trained GCN is then evaluated on a similarly rewired test set. We use a similar network architecture for the GCN as the various graph attention networks described in **Appendix** A.4, with the graph attention layers replaced with graph convolutional layers. We set the number of hidden dimensions in the GCN models to be $F' = 100$.

For the Chameleon dataset, we identified the hyperparameter settings that contributed to the highest validation accuracy for each of the one-layer GAT, GATv2, and Transformer CAR-trained models. We then trained GCN models on graphs that are pruned based on each of these models' attention mechanisms. We also pruned graphs using the counterparts of these models that were trained without CAR and trained another set of GCN models on these pruned graphs. We evaluated the test accuracy of the GCN models when performing this procedure across various attention thresholds (**Figure** 7). We observed that training and evaluating GCN models on pruned graphs contributed to enhanced test accuracy compared to the baseline GCN models that were trained and evaluated on the original graph. Furthermore, we compared GCN models trained and evaluated on CAR-guided pruned graphs against similar GCN models trained and evaluated on graphs pruned without CAR by computing the area under the curve (AUC) associated with the test accuracies at various attention thresholds. Each AUC was calculated as the area below its models' test accuracies line and above the baseline GCN models' test accuracy. CAR-guided graph pruning was associated with higher AUC values across the three graph attention mechanisms, demonstrating the potential for CAR's utility in graph pruning tasks.

**Figure 7:** Test accuracy of models trained and evaluated on graphs rewired using graph attention

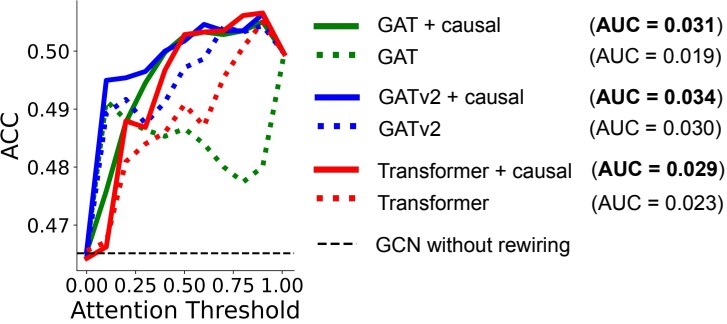

## A.14 ATTENTION INTERPRETABILITY

**Table 11:** Down-weighted citations of "anchor" papers. Each of the cited works that is down-weighted is a well known ML paper.

| Paper Title | Cited Title | Δ |
|---|---|---|
| *Reservoir Computing Hardware with Cellular Automata* | *Adam: A Method for Stochastic Optimization* | -99.87% |
| *Quantization Networks* | *Deep Residual Learning for Image Recognition* | -99.86% |
| *Compressive Hyperspherical Energy Minimization* | *ImageNet Large Scale Visual Recognition Challenge* | -99.75% |

**Table 12:** Upweighted highly relevant citations.

| Paper Title | Cited Title | Δ |
|---|---|---|
| *Generalized Random Gilbert Varshamov Codes* | *Expurgated Random Coding Ensembles Exponents Refinements and Connections* | 7700 % |
| *Sign Language Recognition Generation and Translation: an Interdisciplinary Perspective* | *Swift a SignWriting Improved Fast Transcriber* | 2470 % |
| *Random Beamforming over Quasi-Static and Fading Channels: A Deterministic Equivalent Approach* | *Optimal Selective Feedback Policies for Opportunistic Beamforming* | 536 % |

