# OpenReview forum: "Causally-guided Regularization of Graph Attention improves Generalizability"
_ICLR.cc/2023/Conference — Submitted to ICLR 2023_

### Official Review · Reviewer_J9zG · 2022-10-22

**Confidence:** 4
**Correctness:** 3
**Technical Novelty And Significance:** 2
**Empirical Novelty And Significance:** 2
**Recommendation:** 3

**Clarity, Quality, Novelty And Reproducibility:**

The description of the method is confusing. It is difficult to understand how to implement the method from the current manuscript.
Theoretical proofs and methodological explanations are missing.


**Strength And Weaknesses:**

Strength:
1.	They conduct extensive experiments to demonstrate the effectiveness of their work.
2.	The issue of spurious correlation in GNNs is significant.

Weaknesses:
1. The authors claim that they can solve the problem of spurious correlations. But where is the spurious correlation in their tasks? The authors do not give any detailed descriptions or examples, nor do they show examples of failures of mainstream approaches.
2. Does the example in Figure 1 exist? The authors use the example of Figure 1 to introduce confounders, but it is doubtful whether there is a corresponding phenomenon in the real-world dataset.
3. The authors claim that they can improve ANY attention-based GNNs, but only GAT-like networks are introduced in the experiments. But there exist numerous attention-based methods for GNNs [1-5]. It is questionable then whether it will work on other types of attention mechanisms.
4. For equation (5), the authors only give the regularization term they propose, but do not explain it from a theoretical and causal point of view. For equation (4), the author does not give a detailed definition of $L^p$.
5. How the edge intervention process is implemented, the author does not give a detailed algorithm process or model architecture diagram to explain in detail.
6. For experiments, the author reproduces the statement that they use a public data splitting method. However, compared to the public results, the performance of node classification is questionable, for example, the performance of GAT on Cora is probably around 80-83, based on mainstream packages, such as Pyg or DGL.
7. The authors claim that their method can improve generalization by overcoming spurious correlations, but lack many baseline methods, such as [6].

[1] Gated Graph Sequence Neural Networks, ICLR 2016
[2] How to find your friendly neighborhood: Graph attention design with self-supervision, ICLR 2020
[3] Self-attention graph pooling, ICML 2019
[4] Understanding Attention and Generalization in Graph Neural Networks, NeurIPS 2019
[5] Causal Attention for Interpretable and Generalizable Graph Classification, KDD 2022
[6] Handling Distribution Shifts on Graphs: An Invariance Perspective, ICLR 2022


**Summary Of The Paper:**

This paper claims that existing attention-based GNNs are vulnerable to spurious correlations, thereby hampering the generalizability of the model. To improve the generalization ability of the attention-based GNNs, the authors propose a causal inference approach. Specifically, they use a causally-inspired regularization scheme (CAR).  It can be applied to any attention-based GNN architecture. They conduct experiments on node classification tasks, and find the accuracy improvements and loss reductions.

**Summary Of The Review:**

The authors claim that they can solve the spurious correlation problem, but give no examples of their task or failures of existing methods. The description and theoretical justification of the method in this paper are insufficient.

---

> ### Author Response · Authors · 2022-11-17
> **Point-by-point response**
>
> We appreciate the reviewer’s detailed comments and helpful feedback. Please see above for a broad summary of major revisions and below for detailed point-by-point responses to each question or weakness raised by the reviewer.
>
> 1. Since this and the next point are related, please see our response for the next point.
>
> 2. We agree that systematically defining spurious correlations is a challenge that has been central to many important works related to casual model discovery, such as [6] mentioned below. That said, we respectfully disagree that our work does not relate to spurious correlations in real-world data. We discuss two examples from our paper: task-relevant homophily and down-weighting of self-citations in obgn-arxiv.
>
>     In the “Connection to homophily” section, we discuss how our method could reduce the effects of non-homophilic spurious correlations. We described a label-agreement based measure for assessing the accuracy of attention mechanisms and show that, compared to baseline, our method achieves higher accuracy (Fig 3). We show this systematically with synthetic Cora datasets with varying levels of homophily (Fig. 6). Since spurious connectivity is likely non-homophilic, this indicates our ability to account for such concerns.
>
>     Also consider the examples shown in Table 3 (“Down-weighted self-citations” in obgn-arxiv). This table highlights pairs of publications that have similar-looking abstracts and titles but that are nonetheless labeled as being from different subject areas. Given the differing labels, the strong textual similarity between these pairs of articles is effectively a spurious correlation. Our method is able to downweight these spurious relationships effectively.
>
> 3. We agree with the reviewer about the importance of these alternative attention-based GNN methods, several of which we had referenced in the original paper. We are grateful for the additional references, and have included a discussion of these in our draft.
>
>     The experiments in our paper initially focused on GAT-like networks due to the accessibility of attention weights in the PyTorch Geometric implementations of these models. We also note that some of the methods cited by the reviewer (e.g. [4]) are actually GAT-like in their core implementation. We are eager to assess our method on many of the other attention-based methods, though we do not anticipate that we can cover all of the aforementioned methods given the brief rebuttal period.
>
> 4. We have modified our paper to include a more thorough theoretical justification of our proposed method. Please see our response to Comment 2 from Reviewer iUpK above, which includes more details on how we’ve addressed this issue.
>
>     With regards to L^p in Equation 4, we have included an additional equation in the text introducing this equation to further clarify that L^p represents the loss associated with the prediction. We note that Equation 4 from the original manuscript appears as Equation 1 in the updated manuscript.
> 5. To more clearly describe the edge intervention procedure and how it fits into our framework, we have added pseudocode for the method in the Appendix that details each step of the algorithm. Relatedly, details on the model architecture can be found in Appendix A.4 entitled “Graph Attention Network Architectures and Training”.
> 6. Please see our response to Comment #2 from Reviewer 5nKt above.
> 7. We appreciate the reviewer for pointing us to this important work, which we have now referenced in the revised version of our paper. While we aim to leverage invariance principles like in [6], we note that our method differs in that its primary goal is to improve generalization (by reducing the effect of spurious correlations on graph attention training) without necessarily assuming a distribution shift. In contrast, the explicit goal in [6] is to handle distribution shifts. Furthermore, even among approaches that invoke invariance principles, our work differs in that it uses an intervention-based strategy, while [6] and other related works predominantly leverage data augmentation strategies.

---

### Official Review · Reviewer_iUpK · 2022-10-24

**Confidence:** 4
**Correctness:** 3
**Technical Novelty And Significance:** 2
**Empirical Novelty And Significance:** 2
**Recommendation:** 5

**Clarity, Quality, Novelty And Reproducibility:**

Clarity: Good

Quality: Fair

Novelty: Limited

Reproducibility: Good

**Strength And Weaknesses:**

Strengths:
1. The paper conducts abundant experiments on a broad range of datasets, which shows consistent performance to demonstrate the effectiveness of CAR.
2. Qualitative analyses are provided to suggest that the attention-weight changes produced by CAR are intuitive and interpretable.

Weaknesses:
1. The paper shares a lot of common idea with the work “Causal Attention for Interpretable and Generalizable Graph Classification, KDD22”, but lack discussion and comparison with it.
2. The paper does not provide an analysis and discussion of the causal effect score, which is a very important component of the work. Is it theoretical solid in its current form? And experiments are also required to prove its effectiveness compared to other estimations of the causal effect.
3. Equation 5, p_ij^{n} can be greater than 1 in practice, since removing some noisy edges can also improve the performance. And d(n) >1 will make the value C greater when d(n) increases. However, the author claims that “per-edge attention weights are expected to be smaller for higher-degree nodes”.


**Summary Of The Paper:**

The Paper argues that existing studies to improve the generalizability of attention mechanism in GNN are independent of the actual prediction without regard for the training objective. Thus the paper leverages active interventions on nodes’ neighborhoods by deleting some specific edges to align graph attention training with the causal impact of these interventions on task performance. Experiments on three graph attention architectures across an eight-node classification task show improvements.

**Summary Of The Review:**

The proposed method is effective and easy to follow. However, the novelty needs to justify based on related works. And an analysis and discussion of the causal effect score is also required.

---

> ### Author Response · Authors · 2022-11-17
> **Point-by-point response**
>
> We appreciate the reviewer’s detailed comments and helpful feedback. Please see above for a broad summary of major revisions and below for detailed point-by-point responses to each question or weakness raised by the reviewer.
> 1. We thank the reviewer for drawing additional attention to this work. While we had already acknowledged it as an important reference (“Introduction” and “Related Work”), we now have a more extensive discussion of this work and a direct comparison to it.
>
>     Our method takes a more direct edge-wise intervention approach, is suitable for node-level predictions, and is aimed more at generalization performance than out-of-distribution robustness. While Sui et al. (2022)’s method (termed CAL) and our paper have related motivations, CAL uses perturbations on abstract graph representation to perform causal interventions while we propose a more direct strategy (intervening on the graph edges themselves) that enables causal effects to be computed scalably. In addition, CAL is designed to identify causally attended subgraphs for graph property prediction tasks, while our work primarily focuses on node classification tasks. Furthermore, CAL seeks a model that is robust to interventions, while we measure the effects of an intervention and use that to guide model training. To make these differences clearer, we have provided a more detailed discussion of the relationship between our paper and this work in the “Related Work” section.
>
>     To more directly compare our work and CAL, we also slightly modified the CAL architecture to make it compatible with node prediction tasks (see “Generalization performance” subsection in “Results”, Appendix A.9). This modification only entailed removing the final pooling layer in the CAL model that aggregates all node representations within a graph upstream of a classifier, thereby enabling node-level predictions. Compared to our models, we found that CAL models yielded substantially lower test accuracies.
> 2. We have made extensive revisions to our paper to more comprehensively present the theoretical foundations of our method. While we had previously alluded to a link between our method and the invariance prediction framework, we now formalize this connection in Section 2.2. We describe the relationship between our work and the invariance prediction principle for causal inference, show how our regularization loss is related to analyzing model residuals in the invariant framework, and link this formulation to our causal effect calculation.
>
> 3. The reviewer is correct in that $\rho_{ij}^{n}$ can indeed be greater than 1, but such a scenario would indicate worsened performance rather than improved performance. For $\rho_{ij}^{n} > 1$, the loss associated with the edge intervention is greater than the loss in the absence of the intervention, indicating worsened performance upon intervention. In addition, the reviewer’s point about the removal of noisy edges improving performance is exactly correct and matches our intuition, as the removal of noisy edges would lead to $\rho_{ij}^{n} < 1$ and, thus, a smaller causal effect $c_{ij}^{n}$.
>
>     With regards to the reviewer’s point about d(n), we intended to convey the idea that interventions will likely have smaller effects on higher degree nodes due to the increased likelihood of there being multiple edges that are relevant and beneficial for predictions for such nodes. As such, we included the d(n) term to account for this effect. We have modified our explanation of this node degree-related correction. While d(n) does not strictly fit into a causal invariance hypothesis testing framework, we found that its inclusion empirically led to better generalization performance.

---

### Official Review · Reviewer_5nKt · 2022-10-24

**Confidence:** 3
**Correctness:** 4
**Technical Novelty And Significance:** 3
**Empirical Novelty And Significance:** 2
**Recommendation:** 5

**Clarity, Quality, Novelty And Reproducibility:**

Clarity is good. The presentation is clear and the concepts are easy to follow.

Quality is good. The experiments are extensive and well designed.

The idea to regularize attention coefficients based on edge interventions is interesting, but not very surprising, given prior work that study similar problems. The proposed regularization framework is incremental.

**Strength And Weaknesses:**

Pros:
- The paper is well structured, well motivated and nicely written. The presentation is clear, the idea is interesting, and the results are easy to follow.

Cons/Questions:
- The test accuracies reported in Table 11 in Appendix A.6 are much lower than what are typically achieved by GNNs. For example, the test accuracy for Cora is usually higher than 0.8 for most existing GNN methods with 1-2 layers, but in Table 11 the test accuracy for GAT is only 0.58. Could the authors comment on what differences had caused such a large gap between existing results and the results reported in this work?
- For semi-supervised node classification task, usually only a small fraction (e.g. 5% or less) of nodes are labelled. In this case, does it mean that the regularization loss in Equation 3 is computed only on the training nodes, and consequently only the attention coefficients for the training nodes are regularized? If this was true, then the regularization would have no effect on the test nodes. I think I must have misunderstood something, could the authors help clarify the effect of regularization on attention coefficients for nodes whose labels are unknown?

**Summary Of The Paper:**

This paper proposes a regularization framework for training graph attention networks. The proposed method is based on the idea of performing edge interventions to determine the relative importance of an edge to specific prediction tasks. Extensive experiments show that the new regularization framework helps improve generalization while at the same time makes the attention coefficients more interpretable.

**Summary Of The Review:**

The paper proposes an interesting regularization framework to improve the generalizability and interpretability of graph attention networks. The idea is interesting and the paper is well written. However the overall method development is incremental and I still have some questions regarding the empirical results. I would increase the score if the authors properly address those questions.

---

> ### Author Response · Authors · 2022-11-17
> **Point-by-point response**
>
> We appreciate the reviewer’s detailed comments and helpful feedback. Please see above for a broad summary of major revisions and below for detailed point-by-point responses to each question or weakness raised by the reviewer.
>
>   We would also like to address the reviewer’s concluding remarks that:
>     *The idea to regularize attention coefficients based on edge interventions is interesting, but not very surprising, given prior work that study similar problems. The proposed regularization framework is incremental.*
>
>   While some previous work has indeed looked at causal interventions on graph structures, these approaches have either taken an unsupervised approach, performed interventions on implicit representations, or used causal interventions to re-sample networks for improved out-of-distribution performance. Our goal is more focused: improved generalization on node classification tasks. We argue that our intervention is a more intuitive, direct and scalable approach towards this goal. It can directly be incorporated in a variety of graph attention architectures.
>
> 1. Thank you for the opportunity to clarify this. We sought to assess the effectiveness of CAR as a general approach, and hence evaluated it on a broad range of hyperparameter choices (48 hyperparameter settings for each of 8 datasets x 3 GNN architectures= 1152 total), rather than optimizing hyperparameters for a singular model. Thus the test accuracies reported in Table 11 reflect the results from the full set of hyperparameter combinations (see Section 4.1 and Appendix A.4) and are hence lower than what the single-best setting would presumably yield. On a related note, we also used the standard PyTorch Geometric implementation of the graph attention layers in order to ensure a clean comparison.
>
>     We have now provided additional explanations about our experimental setup in Section 4.1 which clarify that our analysis was intended to demonstrate the broad applicability of CAR across hyperparameter choices.
>
> 2. You are correct that the regularization loss is computed only for the training nodes. However, we note that the attention mechanism consists of learnable parameters that are learned from training data but can then be applied to *all* data (incl. test) (see Table 4 in Appendix A.1 for details of various attention mechanisms). Note that the normalized attention coefficients (i.e. $\alpha_{ij}$) that indicate edge scores are not parameters themselves: they are computed by applying the attention mechanism on the node representations. Thus, regularizing with respect to these attention coefficients during training affects the overall attention mechanism. In turn, the attention mechanism has an effect on all evaluated nodes, not just the training nodes. In particular, improvements in a model’s attention mechanism would  translate into improvements in the model’s ability to generalize to test nodes.
>
>     We have revised the “Causal Attention Regularization Scheme” section to address any confusion related to the role of regularization on the learned attention mechanism.

---

### Official Review · Reviewer_ZGgA · 2022-10-25

**Confidence:** 4
**Correctness:** 3
**Technical Novelty And Significance:** 3
**Empirical Novelty And Significance:** 3
**Recommendation:** 5

**Clarity, Quality, Novelty And Reproducibility:**

Parts of the paper that are related to the algorithm should be more clearly written. The approach is somehow novel but it should be better and more precisely stated and motivated. The experiments should be easily reproducible is the authors eventually make the code available.

**Details Of Ethics Concerns:**

No specific concerns to report.

**Strength And Weaknesses:**

Strength: The proposed idea is relatively intuitive and valid. The paper is generally easy to follow.

Weaknesses: Parts of the paper (e.g., definition of causality, discussion of experimental results) are superficially treated (please refer to the detailed comments below)

**Summary Of The Paper:**

The paper introduces a framework for learning attention weights on the edges of the graph through a graph attention framework. The learned attention is causally-guided, which means that it aims at aligning the attention mechanism with the causal effects of interventions on the graph connectivity. The causal effect of an edge is measured through the impact it has in the final task, thus it is integrated in a loss function of the training phase. The authors show that the proposed framework improves the performance on node classification tasks in standard datasets. Moreover, the learned attention seems to be more interpretable.

**Summary Of The Review:**

Below are a few comments that should be addressed by the authors, and could hopefully improve the quality:

1) The causal attention regularization scheme is not clear. A summary of the algorithm should be provided, or a schematic overview that would help the reader understand how Eq (5) is integrated into (3). For example, $\sigma$ in Eq 5 is not defined. Please make the algorithm as accessible as possible to readers.

2) Is there a reason why the regularisation is applied only to attention networks? Could it be applied to any other architecture as well by adding a regularisation term that represents the auxiliary supervised prediction task that aim to align the weights of the networks with the causal effect of removing specific edges or nodes on the graph?

3) The definition of causality is vague as it captures more correlation of a specific edge with the outcome than causality. Can you comment/clarify that part?

4) How do you define R? It seems that it is an important parameter of your algorithm, and one of the reasons why it does not scale on real graphs.

5) While the method is tested in different datasets, the results are not sufficiently described. Can you comment on the performance of homophily/heterophily datasets? Is there any specific pattern? Does this improved attention scheme help in one or another direction?

6) The neighbor voting regularization scheme is not clear. Please make it precise and clear.

7) The graph pruning approach needs also more details. Why is that method more interesting than classical pruning approaches?

8) While the intuition of the paper is reasonable, there is no clear conclusion on the usefulness of the method. The authors touch upon different aspects such as 1) generalisation 2) interpretability 3) graph pruning, but the results are not strong enough in any of these aspects. The postdoc analysis of the interpretability weight on the ogbn-arxiv is also relatively vague, as some of the statements could be subjective.

9) The authors should include F1 scores in the experiments.

---

> ### Author Response · Authors · 2022-11-17
> **Point-by-point response**
>
> We appreciate the reviewer’s detailed comments and helpful feedback. Please see above for a broad summary of major revisions and below for detailed point-by-point responses to each question or weakness raised by the reviewer.
> 1. We apologize for the lack of clarity regarding the description of our method. We have now added a new section in the Appendix section (Appendix A.1) that includes pseudocode for the method, detailing each step of the algorithm. In addition, we have modified the description of σ from Equation 5 in the “Calculating task-specific causal effects” section to clarify its role in the algorithm.
> 2. We appreciate the reviewer’s suggestion of alternative applications of our regularization method. While our regularization framework could potentially be applicable to non graph-attention settings, we focused on graph attention networks because of their immediate relevance to our primary goal: identifying and downweighting the edges in a graph that are not relevant to a prediction task. The attention formulation is very well suited to identifying task-relevant edges in an interpretable way.
>
>     However, we remain very much interested in exploring extensions of our work, including the possibility of employing similar ideas to networks that don’t explicitly use attention, and we have included a brief description of such future directions in the “Conclusion” section.
> 3. We apologize for the lack of clarity in describing the role of causality in our proposed method. Our approach derives from the invariant prediction formulation for causal inference, in which the set of models under which predictions are invariant to interventions comprise the causal model. We have clarified this throughout the main text and have also added an extensive discussion of the theoretical foundations of our work in Section 2.2.
>
>     Please also refer to our response to Comment #2 from Reviewer iUpK, where we also provide a more detailed explanation of the relationship between causal inference, invariance prediction, and our method.
> 4. We define R to be the number of rounds of interventions to be performed per mini-batch. Note that each round intervenes on all training nodes in a mini-batch in parallel. We respectfully disagree with the reviewer’s comments about our method’s scalability to real graphs. Our approach for performing interventions was designed specifically to enable parallelization of causal effect calculations, which substantially aids in scalability— the runtime of our method is typically less than 2x that of the baseline, non-causal setting. We have provided further clarifications to the relationship between the choice of R and scalability in the “Edge intervention procedure” and “Scalability” sections of our paper.
>
>     We have also added a more detailed exploration of the relationship between this hyperparameter R and the effectiveness of our method in the “Appendix” and “Results” sections. These results show that the choice of R has minimal effect on the accuracy improvement associated with using our method. This may possibly be due to there being a sufficient number of causal effect examples for even the minimum R=1 case, in which we use roughly as many causal effect examples as training examples. Relatedly, for R > 1, we use roughly R causal effect examples per training example.
> 5. We have performed additional experiments to directly assess the effectiveness of our method with respect to homophily. We evaluated synthetic Cora datasets with varying levels of homophily and have included these new results in the “Results” and “Appendix” sections. Consistent with our method's ability to discern relevant edges, its relative outperformance is greater on data with low homophily.
> 6. We agree that this was unclear and have revised the subsection that discusses the neighbor voting scheme, which we renamed “Comparison to homophily-based regularization”. Briefly, this section describes an ablation study: having demonstrated that our method captures task-relevant homophily (Fig 3), we wondered if it did so better than an alternative approach to prioritizing homophily (neighborhood voting).
> 7. We have provided a more detailed description of our graph pruning approach alongside a more comprehensive discussion of its relationship to existing graph rewiring approaches in the Appendix. We highlighted that this approach differs from alternative approaches in that it directly uses local pruning (i.e. edge interventions) to guide the training of graph attention mechanisms that down-weight task-irrelevant edges. In comparison, many methods need to perform whole-graph computations to identify prunable edges, which can be infeasible for very large graphs. In our preliminary analyses we have found our graph pruning strategy to be effective in enhancing the generalizability of GCNs.

---

> > ### Author Response · Authors · 2022-11-17
> > **Point-by-point response (continued)**
> >
> > 8. We are sorry that this wasn’t clearer. We have now updated the “Conclusion” to more clearly set out our innovations. Our paper has a focused goal: improved generalization by better training of graph attention networks. Unlike other causal approaches, we are not seeking out-of-distribution robustness by re-sampling from the network. Our core intuition is direct and effective: causal edge interventions help tune attention to better capture task-relevant homophily in the presence of noise and confounders. We do so with a scalable approach that consistently outperforms the baseline. We prioritized the reporting of the consistency of our performance across a variety of settings, rather than any specific SOTA results: we report 1152 datapoints comparing our method and the baseline: 3 architectures x 8 datasets x 48 hyperparameter settings.
> >
> >     We believe the interpretability of our model is an important side-benefit and a way to confirm that our attention tuning is indeed happening in the correct way. The graph pruning results are preliminary analysis and we hope to explore those more in future work.
> > 9. We appreciate the reviewer’s suggestion to consider other performance metrics. We chose to use accuracy as an evaluation metric as this is in line with standard literature practice in the context of these datasets. The prediction task for all of the benchmark datasets used in our paper is multi-class classification, for which performance can be readily summarized using accuracy.
> >
> >     While we agree that F1 scores can be a helpful metric in many cases, we find them to be less suitable in a multi-class setting such as ours since a sequence of somewhat arbitrary decisions are needed to produce a single F1 score for each model: 1) an F1 score would need to be computed for each class separately, 2) computing this F1 score requires manually specifying a threshold for each class, and 3) a summarization operation (e.g., average, weighted average) needs to be applied across all of the per-class F1 scores to produce one score per model.

---

### Author Response · Authors · 2022-11-08
**Preliminary response to reviewers and plan of action**

We thank the reviewers for their valuable feedback. We are working to address the various points raised by each reviewer and wanted to share a preliminary note describing the key improvements that we plan to make in our revised paper. We plan to…

1. Provide a more thorough theoretical justification of our proposed method. We had briefly alluded to the connection between our method and the invariance prediction framework for causal inference and we will expand on this connection.
2. Perform a more systematic analysis of the relationship between spurious correlations, invariance, and the effectiveness of our method.
3. Compare our proposed method against related approaches that employ concepts from causal theory for graph attention training.
4. Provide a clearer description of our experimental setup, especially with respect to hyperparameter choices. We aimed to demonstrate the consistency of our proposed method’s effectiveness across a wide range of hyperparameter choices rather than fixating on a limited set of models with optimized hyperparameters.
5. Perform a more systematic interpretability analysis of the attention weights for the ogbn-arxiv dataset.

We welcome any feedback on this outline of planned revisions.

---

### Author Response · Authors · 2022-11-17
**Summary of Revisions**

**Updates:** We have edited this note after the submission of our revised manuscript to highlight additional evaluations and theoretical justifications.

We appreciate the reviewers for taking the time to provide detailed and valuable feedback on our paper. We have made substantial revisions to address the reviewers’ concerns and believe these modifications have greatly improved our paper. While we put the finishing touches on our updated manuscript, we wanted to share our responses in case the reviewers have additional feedback that can be quickly incorporated in the next two days.

We broadly summarize the major revisions here and have included detailed point-by-point responses to each review below.
- Extensive theoretical justification of our proposed method, which we relate formally to the invariance prediction formulation of causal inference.
- Evaluation on synthetic Cora datasets to further demonstrate the relationship between our method's performance and homophily.
- More comprehensive comparisons of our proposed method with related work (incl., Sui et al, KDD 2022) and included additional citations of important related work.
- More detailed descriptions of our proposed algorithm and added a new section in the Appendix outlining our implementation of the algorithm.
- Clarification of the relationship between spurious correlations and our experimental results.
- Clarification of experimental setup and analyses.

---

### Decision · Program_Chairs · 2023-01-20

**Decision:**

Reject

**Justification For Why Not Higher Score:**

Limited novelty and inconclusive as to whether it will provide benefits over the current state of the art.

**Justification For Why Not Lower Score:**

N/A

**Metareview: Summary, Strengths And Weaknesses:**

(a) This paper introduces a regularization scheme for training graph attention networks that aligns graph attention with the causal impact of interventions on graph edges. With extensive experiments on eight node classification tasks and three graph attention architectures,  the proposed regularization scheme is shown to help improve model generalization and interpretability compared to without the regularization.
(b) The proposed idea is intuitive and improves performance when applied to several graph attention models.
(c) As noted by several reviewers, the main idea of this work is somewhat incremental, limited in novelty. The main evaluation compares training with and without the proposed regularization term. While this does show improved performance, it is not clear whether the approach will actually improve upon the state-of-the-art as the reported results are over large range of (suboptimal) parameter settings.

**Summary Of Ac-Reviewer Meeting:**

N/A